

# Aero-elastic Wind Turbine Design with Active Flaps for AEP Maximization

Michael K. McWilliam[1], Thanasis K. Barlas[1], Helge A. Madsen[1], and Frederik Zahle[1]

[1]DTU Wind Energy, Frederiksborgvej 399, 4000 Roskilde, Denmark

*Correspondence to:* Michael K. McWilliam (mimc@dtu.dk)

**Abstract.**

In optimal wind turbine design, there is a compromise between maximizing the energy producing forces and minimizing the absolute peak loads carried by the structures. Active flaps are an attractive strategy because they give engineers greater freedom to vary the aerodynamic forces under any condition. Flaps can be used in a variety of different ways (*i.e.* reducing fatigue, peak loads *etc.*), however this article focuses on how quasi-static actuation as a function of mean wind speed can be used for Annual Energy Production (AEP) maximization. Numerical design optimization of the DTU $10MW$ Reference Wind Turbine (RWT), with the HAWTOpt2 framework, was used to both find the optimal flap control strategy and the optimal turbine designs. The research shows that active flaps can provide a 1% gain in AEP for aero-structurally optimized blades in both add-on (*i.e.* the flap is added after the blade is designed) and integrated (*i.e.* the blade design and flap angle is optimized together) solutions. The results show that flaps are complementary to passive load alleviation because they provide high-order alleviation, where passive strategies only provide linear alleviation with respect to average wind speed. However, the changing loading from the flaps further complicates the design of torsionally active blades, thus, integrated design methods are needed to design these systems.

## 1 Introduction

The size of wind turbines has been increasing rapidly over the past years. Rotors of more than $160m$ in diameter are already commercially available. By focusing on lowering the cost per kWh, new trends and technological improvements have been primary targets of research and development. One main focus (among others) is on developing new technologies [Madsen et al. (2014)] which are capable of considerably reducing the loads and increasing Annual Energy Production (AEP) for wind turbines.

New concepts for dynamic load reduction are focusing on both a faster and detailed load control, compared to existing individual blade pitch control, by utilizing active aerodynamic control devices distributed along the blade span [Barlas and Kuik (2010)]. Such concepts are generally referred to as smart rotor control, a term used in rotorcraft research. Over the past years, these concepts have been investigated for wind turbine applications, in terms of conceptual and aero-elastic analysis, small scale wind tunnel experiments, and recently field testing [van Wingerden et al. (2011); Castaignet et al. (2014)]. For a review of the state-of-the-art in the topic, the reader is referred to [Barlas and Kuik (2010)].



So far, results from numerical and experimental analysis, mostly focusing on trailing edge flaps, have shown a considerable potential in fatigue load reduction [Barlas (2011); Bergami (2013)]. Existing work has focused on application of active flaps on existing blade designs [Barlas et al. (2016a); Madsen et al. (2015)], implicitly showing the potential for reduction of cost of energy. Only recently have researchers started to investigate the potential of an optimized blade design integrating the

use of active flaps. Barlas *et. al.* [Barlas et al. (2016b)] explored optimal blade design when flaps are integrated for fatigue load reduction. This work only looked at one flap control strategy, so there remains questions on the impact of other control strategies.

Rotor design is an inherently multi-disciplinary design problem due to the complex coupling between aerodynamics, structural dynamics and control. To properly assess the impact of new technology on rotor design, one must consider all disciplines

simultaneously. For example, load reduction technology is typically used to increase AEP by increasing the rotor diameter. Multi-disciplinary Design Optimization (MDO) can generate aero-elastically optimized blade designs by considering all the disciplines in the optimization simultaneously. In the recent years several MDO frameworks have been developed to perform wind turbine multi-disciplinary optimization design (Bottasso et al. (2012); Ashuri et al. (2014); Merz (2015a, b); Fischer et al. (2014); Ning et al. (2014); McWilliam (2015)). In this work an optimization framework called HAWTOpt2 [Zahle et al.

(2016)] is utilized which enables concurrent optimization of the structure and outer shape of a wind turbine blade. This tool builds on the experience gained with the HAWTOpt code [Fuglsang and Madsen (1999)] but is otherwise a completely new code-base written in the Python programming language.

An example layout of the blade planform incorporating active trailing edge flap sections in shown in Figure 1a, along with a detailed example of a flap in figure 1b.

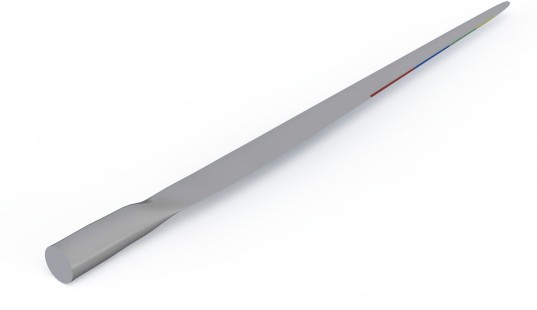 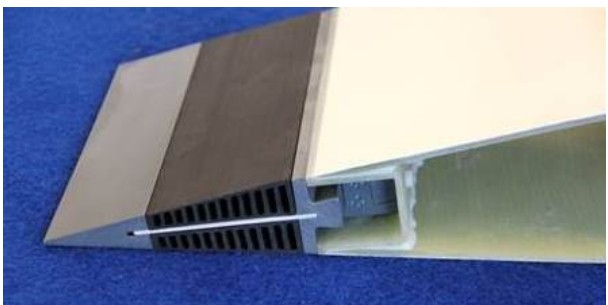

(a) Blade with Flaps                                     (b) Example Flap System

**Figure 1.** Active Flaps on a Wind Turbine Blade

The overall idea is that between 65% and 95% of the blade span, the trailing 10% of the profile will be replaced with an active flap [Madsen et al. (2014)]. The active flaps have a constant chord as they are manufactured in an extrusion process. The blades can be manufactured such that the flap is applied as an add-on to an existing blade or fully integrated into the design





from the start. This paper explored how flaps would affect the design both as an "add-on" and completely integrated in the design. The method of actuation and the structural details of the flap are not included in this analysis.

The overall objective of the paper is to apply the developed aero-elastic optimization framework on the design of a blade, incorporating active trailing edge flaps. There is sufficient freedom in the design variables that the optimization can also incorporate bend-twist coupling. This will demonstrate how both passive and active load alleviation can be combined. The added value of the optimized 'smart blade' design is thus shown through a comparison to an aero-elastically optimized design with no flaps.

## 2  Analysis Methods

MDO was used to investigate how active flaps could improve performance and affect design. The optimization framework is described in section 2.1. The parameterization of the blade is described in section 2.2. The flap configuration and the DTU $10MW$ Reference Wind Turbine (RWT) used in this study is described in section 2.3. Finally, the details about the optimization are given in section 2.4.

### 2.1  Optimization Framework

The HAWTOpt2 optimization framework [Zahle et al. (2016)] is a MDO tool developed by DTU Wind Energy based on OpenMDAO [Gray et al. (2013)]. The tool is a python based code that couples Finite Element Method (FEM) cross section analysis tool BEam Cross section Analysis Software (BECAS) [Blasques et al. (2016)] with the aero-elastic tools HAWC2 [Larsen and Hansen (2014)] and HAWCStab2 [Hansen (2004)] to conduct aero-elastic analysis of wind turbine rotors. The work-flow is shown graphically in figure 2. The analysis is coupled to the interior point optimization algorithm IPOPT [Wächter and Biegler (2006)].

The framework can simultaneously optimize the internal structure, the control strategy and the planform of the rotor. The loads are evaluated with reduced Design Load Cases (DLCs) [Pavese et al. (2016)] which is a simplification of the full set of DLCs defined in [Hansen et al. (2015)]. From this, various load constraints (*e.g* root flap-wise bending moment, tower top thrust, *etc.*) and tip deflection constraints can be evaluated. An envelope of bending loads is generated for the whole blade and then passed to BECAS to evaluate material failure constraints. The framework evaluates the fatigue damage with a frequency domain model from HAWCStab2 [Tibaldi et al. (2015)]. More details on the framework are given in [Zahle et al. (2016)].

### 2.2  Blade Parameterization

The blade planform is described in terms of distributions of chord, twist, relative thickness and pitch axis aft leading edge, the latter being the distance between the leading edge and the blade axis. The lofted shape of the blade is generated based on interpolation of a family of airfoils with different relative thicknesses.

The internal structure is defined from a number of regions that each cover a fraction of the cross-sections along the blade. Each region consists of a number of materials that are placed according to a certain stacking sequence. Figure 3 shows a cross



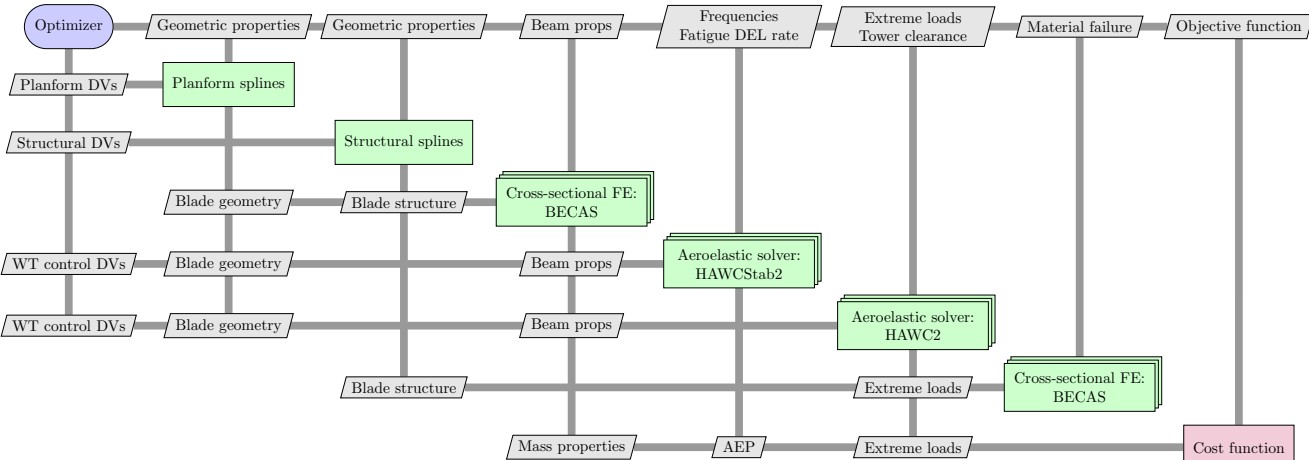

**Figure 2.** HAWTOpt2 Work-flow

section in which the region division points (DPs) are indicated. The DP curves are described by a smooth spline as function of span that takes values between $s$=-1 and $s$=1, where $s$=-1 is located at the pressure side trailing edge, $s$=0 is at the leading edge, and $s$=1 is located at the trailing edge suction side. Shear webs are associated to two specific DPs on the pressure and suction side, respectively, and will move according to these points. The composite layup is likewise described by a series of smooth

5 splines describing the thicknesses of individual layers. For more details on the parameterization see [Dykes et al. (2017)].

To accomodate a flap, the blade is manufactured with a flat back for the flap to be mounted upon. In this study, the presence of the flap was ignored in the structural analysis. However, previous work has shown that the flat back has structural benefits in wind turbine design [Barlas et al. (2016b)].

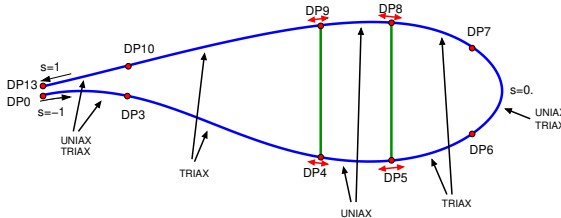

**Figure 3.** Cross-Section Definition





## 2.3 Reference Wind Turbine and Flap Description

The design study is based on the DTU $10MW$ RWT [Bak et al. (2013)]. The RWT is meant to be a $10MW$ off-shore, pitch regulated, wind turbine with conventional glass fiber rotor blades. The aerodynamic design is based on maximizing the $C_P$, so it is not aero-structurally optimal. The loads of this design were used as the constraints for the subsequent designs, so the

5    optimization results here did not increase the loads. Some of the details of this design are shown in table 1, for more details refer to [Bak et al. (2013)].

**Table 1.** Summary of DTU $10MW$ RWT

| | |
|---|---|
| Rated Power | $10MW$ |
| Rotor Diameter | $178.3m$ |
| Rated rotor speed | $9.6rpm$ |
| Rated wind speed | $11.4m/s$ |
| Cut-in, cut-out wind speed | $4m/s, 25m/s$ |
| Gear box ratio | $50.0$ |
| Pitch Rate Limit | $10°/s$ |

    Figure 4 shows the placement of the flap used in this design study. Table 2 shows some of the performance details of this flap. A single set of flaps was used on the rotor between 65% and 95% of the span. In the optimization, the length of the blade is allowed to vary, however the relative span-wise placement of the flaps remains constant. Multiple sets of flaps along the blade

10    could provide finer control of the loading along the span, however this was not considered in this work. The flap deflection was based on the wind speed. The wind speed sensor signal was passed through a low-pass filter. Thus, the flap deflection was quasi-static.

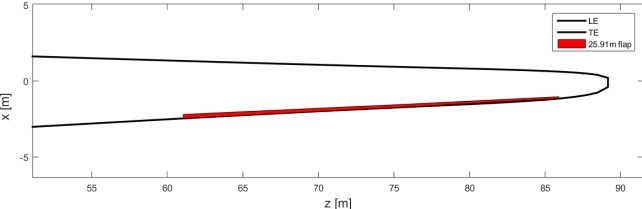

**Figure 4.** Flap Placement



**Table 2.** Summary of Flap Configuration

| | |
|---|---:|
| Chord-wise extension | 10% |
| Deflection angle limits | $\pm 15°$ |
| Span-wise length | $25.9m$ (30% blade length) |
| Span-wise location | $59.59m - 85.80m$ (from blade root) |
| Airfoil | FFA-W3-241 |
| Max $\Delta C_l$ | 0.6 |
| Deflection rate limit | $100°/s$ |
| Actuator time constant | $100ms$ |

## 2.4    Optimization Problem Statement

The optimization is based on solving the problem given in (1). The optimization is essentially a maximization of AEP without increasing the loads on the platform. This is a common industrial industrial design problem where a manufacturer want to develop a new set of blades for a platform already in existence.

$$
\begin{aligned}
\underset{\mathbf{x}_p, \mathbf{x}_s, \mathbf{x}_{oper}}{\text{minimize}} \quad & f(\{\mathbf{x}_p, \mathbf{x}_s\ \mathbf{x}_{oper}, \mathbf{p}, w) \\
\text{subject to} \quad & \mathbf{g}(\mathbf{x}_p) \leq \mathbf{0}, \\
& \mathbf{h_g}(\mathbf{x}_s) \leq \mathbf{0}, \\
& \mathbf{h_s}(\mathbf{x}_s) \leq \mathbf{0}, \\
& \mathbf{k}(\{\mathbf{x}_p, \mathbf{x}_s\}) \leq \mathbf{0}
\end{aligned}
\tag{1}
$$

Where:

$$
f(\{\mathbf{x}_p, \mathbf{x}_s, \mathbf{x}_{oper}\}, \mathbf{p}) = \frac{AEP(\{\mathbf{0}, \mathbf{0}, \mathbf{0}\}, \mathbf{p})}{AEP(\{\mathbf{x}_p, \mathbf{x}_s, \mathbf{x}_{oper}\}, \mathbf{p})}
\tag{2}
$$

The design variables used in this optimization are shown in table 3. They include a mix of planform, internal structure and control design variables. The critical design variable is the blade length. Load reduction strategies are used to allow the rotor to increase in size without increasing the total loads.

The constraints used in the problem are shown in table 4.

## 3    Results

To understand the following results it is helpful to understand how an aero-elastically optimal blade differs from a $C_P$ optimized blade. Overall, aero-elastic optimization reduces the distributed loading so it has room to increase the power with a larger



**Table 3.** Summary of Design Variables

| Parameter | # of DVs | Comment |
|---|---|---|
| Chord | 6 | - |
| Twist | 5 | Root twist fixed |
| Relative thickness | 3 | Root and tip relative thickness fixed |
| Blade prebend | 4 | - |
| Blade pre-cone | 1 | - |
| Blade length | 1 | - |
| Tip-speed ratio | 1 | - |
| Trailing edge uniax | 2 | Pressure/suction side |
| Trailing edge triax | 2 | Pressure/suction side |
| Trailing panel triax | 2 | Pressure/suction side |
| Spar cap uniax | 4 | Pressure/suction side |
| Leading panel triax | 2 | Pressure/suction side |
| Leading edge uniax | 2 | Pressure/suction side |
| Leading edge triax | 2 | Pressure/suction side |
| DP4 | 5 | Pressure side spar cap position/rear web attachment |
| DP5 | 5 | Pressure side spar cap position/front web attachment |
| DP8 | 5 | Suction side spar cap position/front web attachment |
| DP9 | 5 | Suction side spar cap position/rear web attachment |
| Flap Angle | 5 | Flap angle at 5 wind speeds |
| **Total** | 65 | |

swept area. This can be achieved by simply reducing the thrust loading, because near peak $C_P$ it will only cause relatively small reductions in the $C_P$. So aero-elastically optimal blades typically have lower loading, lower $C_P$ and larger swept area. Additional improvements can be achieved by changing the blade configuration under conditions where peak loads occur (*i.e.* at rated wind speed). This is referred to as load alleviation, examples are bend-twist coupling, collective pitch for peak shaving
5  and other load alleviation strategies. Flaps are just one example of load alleviation strategies, however they can also be used in other conditions to extract more AEP by increasing the loads.

The analysis is based on applying the HAWTOpt2 frame work to solve the problem in section 2.4 and then comparing the results. In this work four different optimized designs were generated to evaluate the performance of the flaps:

**DTU 10MW RWT with Flaps** This is the original DTU $10MW$ RWT with flaps attached. Only the flap deflection was
10      varied in the design optimization. This solution represents an flap add-on for a $C_P$ optimized design.

**Baseline** This is a load neutral aero-elastic design optimization of the original DTU $10MW$ RWT without flaps. In this optimization, no flaps were used. This design represents the best rotor performance that can be achieved without flaps.



**Table 4.** Summary of Optimization Constraints

| Constraint | Value | Comment |
|---|---|---|
| max(chord) | $< 6.2$ m | Maximum chord limited for transport. |
| max(prebend) | $< 6.2$ m | Maximum prebend limited for transport. |
| max(rotor cone angle) | $> -5$ deg | - |
| min(relative thickness) | $> 0.24$ | Same airfoil series as used on the DTU 10MW RWT. |
| min(material thickness) | $> 0.0$ | Ensure FFD splines do not produce negative thickness. |
| $t/w_{sparcap}$ | $> 0.08$ | Basic constraint to avoid spar cap buckling. |
| min(tip tower distance) | $>$ ref value | DLC1.3 operational tip deflection cannot exceed that of the DTU 10MW RWT. |
| Blade root flap-wise moments (MxBR) | $<$ ref value | DLB loads cannot exceed starting point. |
| Blade root edgewise moments (MyBR) | $<$ ref value | DLB loads cannot exceed starting point. |
| Tower bottom fore-aft moments (MxTB) | $<$ ref value | DLB loads cannot exceed that starting point. |
| Rotor torque | $<$ ref value | Ensure that the rotational speed is high enough below rated to not exceed generator maximum torque. |
| Blade mass | $< 1.01$ * ref value | Limit increase in blade mass to maintain equivalent production costs. |
| Blade mass moment | $< 1.01$ * ref value | Limit increase in blade mass moment to minimize edgewise fatigue. |
| Lift coefficient @ $r/R = [0.5-1.]$ | $< 1.35$ | Limit operational lift coefficient to avoid stall for turbulent inflow conditions. |
| Ultimate strain criteria | $< 1.0$ | Aggregated material failure in each section for 12 load cases. |
| Flap Angle | $-15.0 \geq \beta \geq 15.0$ | Flap angle stays within 15 degree bounds |





**Baseline with Flaps** Here, flaps were attached to the rotor design from the baseline optimization. In this optimization only the flap deflection was optimized, the rest of the rotor design is identical to the original baseline optimization. This represents an add-on solution for an aero-elastically optimized rotor.

**Co-Design** This is a load neutral aero-elastic design optimization of the original DTU $10MW$ RWT with flaps. In this opti-
mization both the flaps and the rotor design were optimized simultaneously. This design represents an fully integrated design where the rotor is designed for flaps.

Table 5 shows some of the results from the optimization studies. For aero-elastically optimized blades, the flaps can provide a 1% improvement in AEP, where as on a $C_P$ optimized design they can only give a 0.51% improvement. Aero-elastic optimization reduces the loading, this provides the flap with more room to generate more power through increased loading.
This indicates that flaps may provide larger benefits for rotors with lower specific power. The similar performance between the 'baseline with a flap' and co-design, indicates that for this case there is little benefit designing both the blade and the flap deflection simultaneously. Furthermore, add-on solutions should be just as effective as integrated designs.

**Table 5.** Summary of Optimization Results

| Wind Turbine Design | Gain in AEP [%] | Blade Length [m] | Tip Speed Ratio [-] |
|---|---|---|---|
| DTU $10MW$ RWT with Flaps | 0.51 | 86.37 | 7.5 |
| Baseline | 14.65 | 101.25 | 9.12 |
| Baseline with Flaps | 15.68 | 101.25 | 9.12 |
| Co-Design | 15.69 | 101.69 | 8.59 |
| Co-Design without Flaps | 13.93 | 101.69 | 8.59 |

The fact that the base-line with flaps achieved nearly the same performance as the co-design was not expected. Typically, one expects better designs with more design freedom. Further investigation is needed to determine the reasons why. There are
many possibilities: the optimizations may not have been completely converged; there are multiple local minimum solutions; the design space relatively flat, where multiple configurations give similar performance.

Both the aero-elastically optimized blades achieved much longer blade lengths and a faster tip speed ratio. The Co-Design achieved a $44cm$ longer blade than the baseline, so it is getting even more AEP from swept area. The tip speed ratio for the co-design is lower than the baseline. There are a couple possible reasons, the flap actuation could be increasing the profile
drag, thus increasing the aerodynamic penalty of higher tip speeds. A faster tip speed can cause rated rotor speed to occur before peak loads, once this occurs the tip speed ratio drops and the rotor can shed loads. This is a peak shaving strategy that is commonly exploited in HAWTOpt2 results. Thus, a second reason a slower tip speed ratio occurs in the Co-Design, is that the optimization prefers flap actuation over reduced tip speed ratios, for peak shaving.

Figure 5 shows the planform for both the baseline and co-design. The two aero-elastically optimal blades both appear to
have a more aggressive twist distribution (*i.e.* twisted towards higher loading and stall). However, it will be shown later that these blades undergo increased torsional deflection and need the aggressive twist to compensate.





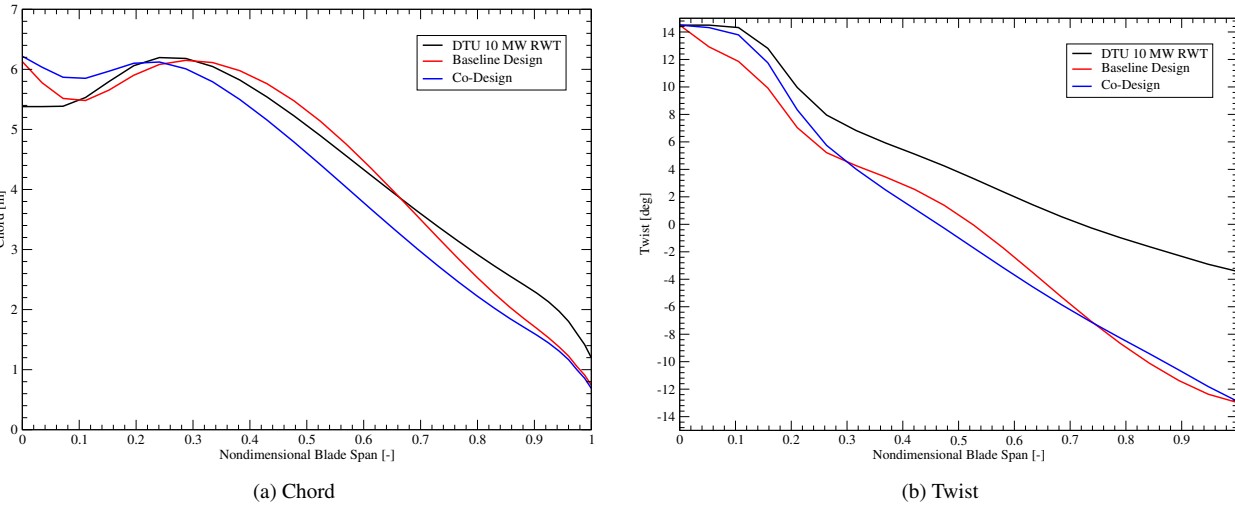

(a) Chord                (b) Twist

**Figure 5.** Rotor Planform Design

Both aero-elastic designs had lower solidity towards the tip. In the co-design, the presence of the flaps in the optimization leads to a smaller solidity and slightly more aggressive twist distribution further inboard. There is an aero-elastic response to the flap deflection, where it will also deform the blade. One may think that the different inboard configuration is compensating for this deformation, however the results will show that is is enhancing the aero-elastic effect of the flap.

Figure 6 shows the optimal flap deflection schedule for the 3 cases with a flap. In all cases the flap deflection is relatively small at low wind speeds because here the turbine is already operating at high $C_T$ values and can generate more power with lower loading. In the constant $C_P$ tracking region, the flap is able to harvest more power with higher loading. Then approaching rated conditions, the flap angle decreases to attenuate the peak loads. The baseline flap schedule is similar to the original DTU $10MW$ RWT, except the lower loading affords more aggressive flap deflections to extract more power. In the co-design the optimization is developing a blade design that relies on much larger flap deflection angles.

Figure 7a shows the power of the optimized designs, the effect of the flap is shown in figure 7b by subtracting the power of the corresponding non-flap blade performance. The results show that the flap adds power in two distinct regions, the first between $4$ and $5m/s$ where the turbine is operating at constant speed. Here the flap is reducing the loading to reduce the high $C_T$. Then between $8$ and $10m/s$ the flap is able to extract more power by increasing the loading.

Figure 8 shows the thrust and coefficient of thrust ($C_T = 2T/(\rho A V^2)$) from steady state simulations of the optimized rotor. It is clear that peak loads occur at $11m/s$. The $C_T$ for all the flap equipped rotors experience a shaper drop between $10m/s$ and $11m/s$, this is evidence that the flaps are being used for peak shaving. Comparing the $C_T$ between the two co-design simulations, it is clear that the optimization is using the flaps more for peak shaving, than in the baseline optimization.

Passive load alleviation is an important innovation for blade design. The HAWTOpt2 framework will produce aero-elastic tailored blades that will passively torsion to shed loads [Zahle et al. (2016)]. This work looked at how passive load alleviation

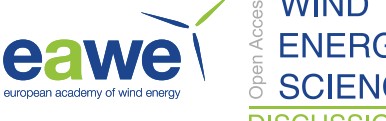

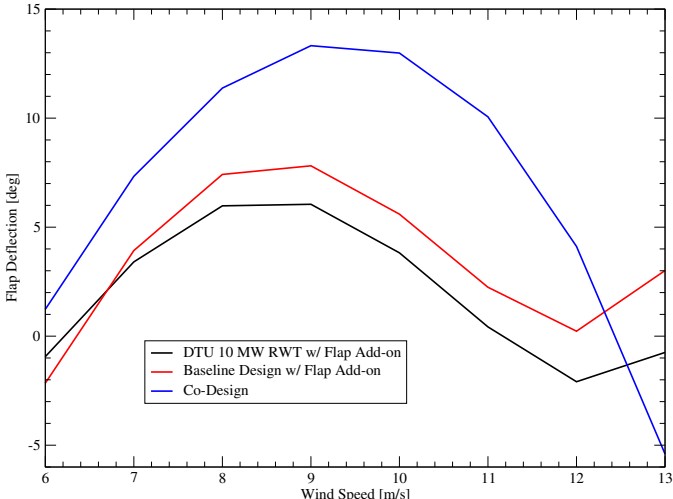

**Figure 6.** Flap Deflection

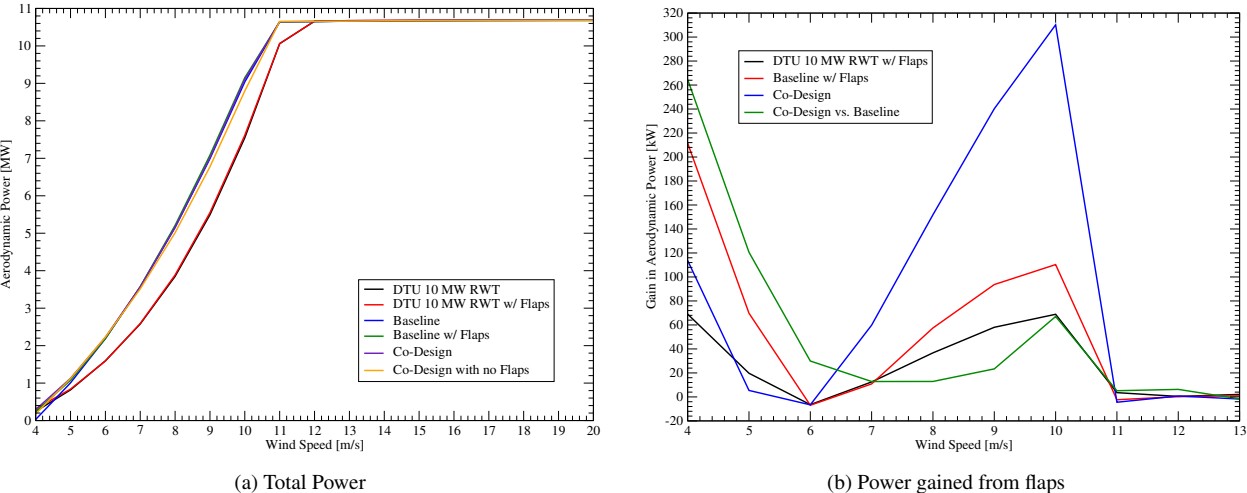

          (a) Total Power                                   (b) Power gained from flaps

**Figure 7.** Aerodynamic Power

could be combined with flaps. Figure 9 shows the torsional deformation at the tip over a range of wind-speeds in steady uniform conditions. In the baseline optimization, the aero-elastic tailoring is optimized without flaps, in the co-design the optimization can vary both the flap angles and the blade design. Comparing these two designs it is clear that both blades had similar amounts of torsional deformation, thus, active flaps complement passive load alleviation strategies. The flap actuation in figure 6 show

5   high order actuation with increasing loads, where as passive strategies can only achieve linear load alleviation.

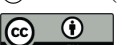


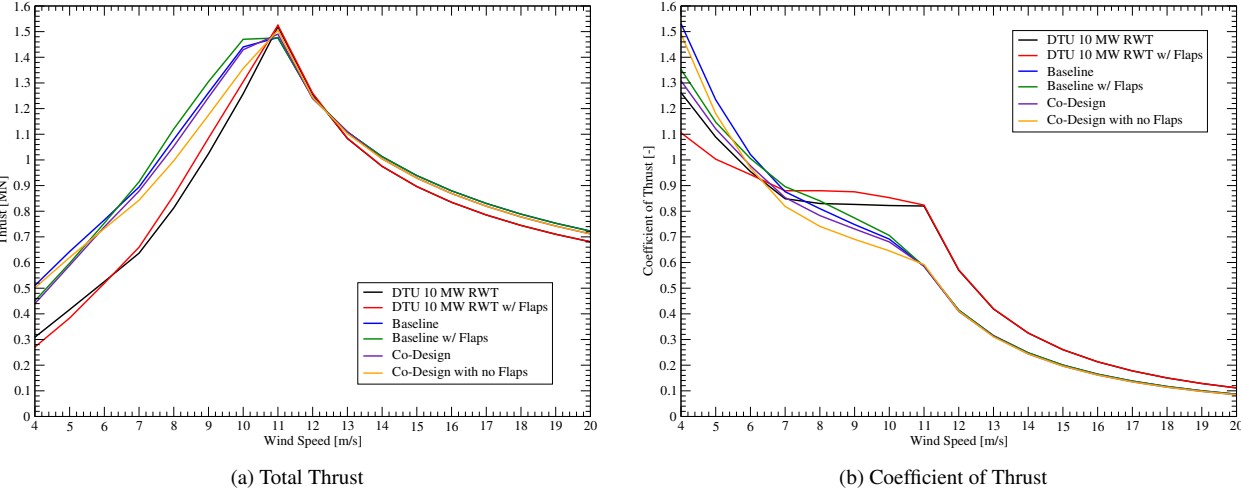

(a) Total Thrust  (b) Coefficient of Thrust

**Figure 8.** Aerodynamic Thrust

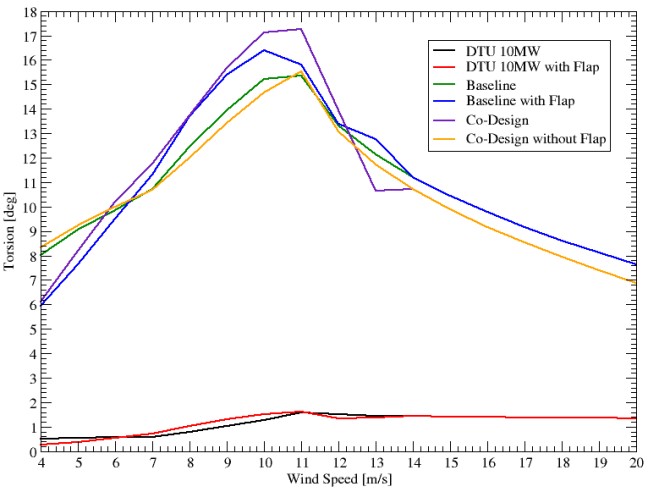

**Figure 9.** Blade Torsion

The results also show that flaps cause additional torsional deformation, because of increased aerodynamic moments on the blade. This suggests that the active load alleviation needs to be designed in-conjunction with the flaps.

Figure 10 shows the distributed aerodynamic forces for the DTU $10MW$ RWT with and without the flaps. The loading caused by the flaps is clear in the $y$ direction (*i.e.* normal to the chord), yet it is quite small in the $x$ direction. In this design the effect of the flaps can be super-imposed on the original aerodynamic loading.

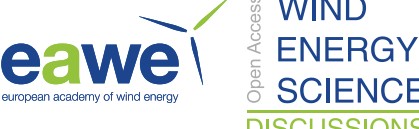



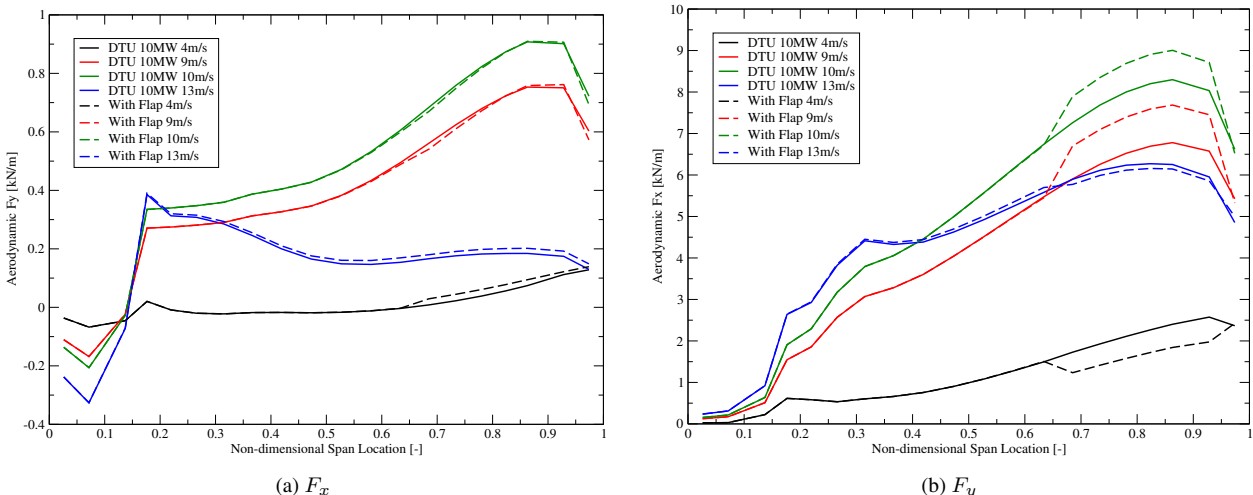

(a) $F_x$            (b) $F_y$

**Figure 10.** DTU $10MW$ RWT Aerodynamic Blade Forces

Figure 11 shows the distributed aerodynamic forces for the baseline rotor, with and without flaps. The loading here is more complicated than in figure 10 because the blade is so much more torsionally flexible. The increased moment caused by the flaps is twisting the blades, this is also seen in the loading in-board of the flaps. This is evidence that the flaps can also redistribute the loads along the blade through elastic torsion of the blade.

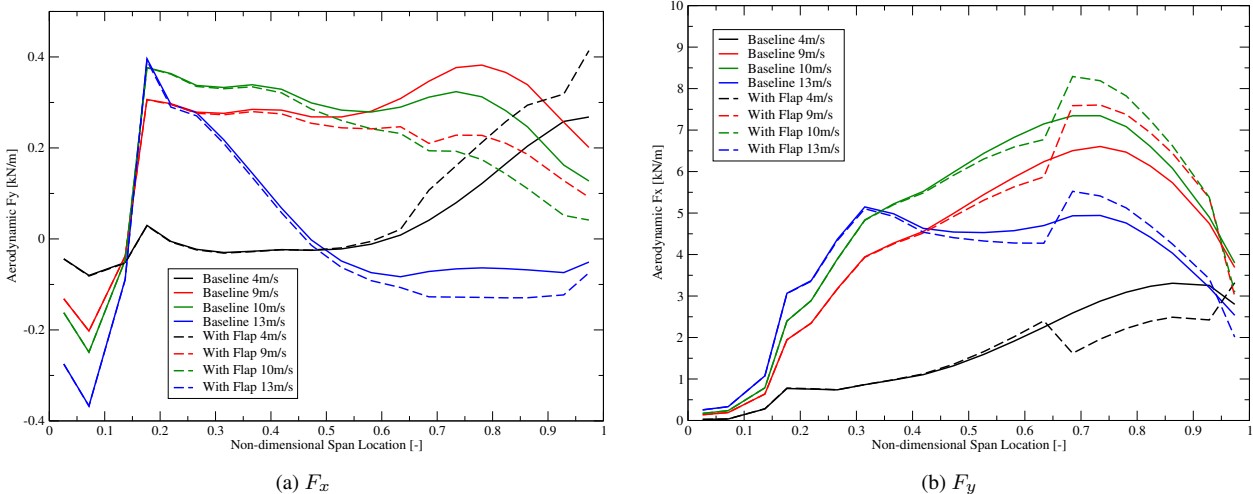

(a) $F_x$            (b) $F_y$

**Figure 11.** Baseline Aerodynamic Blade Forces

5      Figure 12 compares the co-design with flaps to the baseline without flaps. This highlights how including flaps in the optimization leads to different behavior in the final design. In figure 12b the design has a much greater redistribution of loading

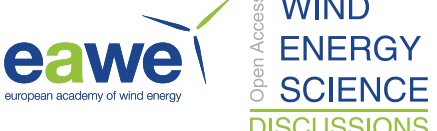

along the blade compared to figure 11. Then in figure 12a the co-design is able to increase the overall driving forces in areas where the flaps are not installed. Thus, an integrated design approach is able to better exploit the different ways the flaps can be used to extract power.

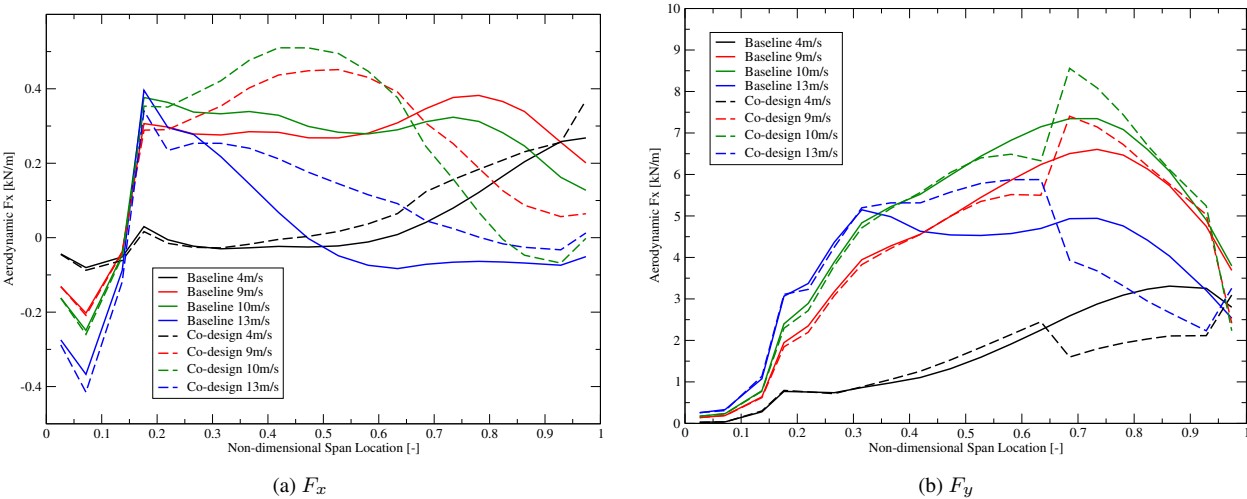

(a) $F_x$          (b) $F_y$

**Figure 12.** Co-Design vs. Baseline Aerodynamic Blade Forces

The figures 11 and 12 shows that the effect of flaps on aero-elastically optimal wind turbine blades is a very complicated. The
5  flaps can change the loading both in the location where they are installed, but also further inboard due changing aerodynamic moment and the torsional flexibility. For these blades, simply super-imposing additional forces from the flaps onto the the original blade loading would not properly capture the true effect of the flaps. The greatest benefit from flaps occurs for aero-elastically optimal blades, yet, the flap control strategies need to be designed with full aero-elastic analysis to accurately predict the true response. This highlights the importance of integrated design tools for advanced rotor design.



## 4  Conclusions

This research investigated how flaps, operated in quasi-static manner, could be used to increase the AEP of a wind turbine. MDO was used to optimize the flap angle and multiple blade designs. For $C_P$ optimized rotor designs, the loading is already quite high, so the flaps could only achieve a $0.51\%$ improvement in AEP. Aero-elastically optimal blades typically have lighter
loading so flaps can be better exploited to extract more energy, giving a $1\%$ improvement in AEP.

The flaps would extract extra energy at very low wind speeds by reducing the loading. They would then extract more energy in the $C_P$ tracking region by increasing the loading. As the wind turbine approached peak loads, the flaps angle would decrease to shave the peak loads. This meant that the flaps would be actuated in a quadratic or higher order actuation schedule. This shows that flaps are complementary with other passive load alleviation strategies that only produce a linear load alleviation.

Flaps introduce a complicated aero-elastic response in aero-elastically optimal blades because they are much more flexible in torsion and flap deflection increases the aerodynamic moment forces. The results shows the flap deflection will twist the blade inboard of the flaps, changing the forces there as well. In the co-design, the optimization was able to find configurations where the flaps contributed more to driving forces inboard of the flaps. Thus, optimal rotor design with flaps cannot be achieved by simply super-imposing the additional loading of the flaps. Thus, full aero-elastic simulations and integrated design techniques
are required to develop these smart rotors.

### Acknowledgement

The work was funded by the Danish Development and Demonstration Program (EUDP) under contract J.nr. 64015-0069, for the research project "INDUFLAP2 - Full scale demonstration of an active flap system for wind turbines."



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
