# Peer review of "Aero-elastic Wind Turbine Design with Active Flaps for AEP Maximization"

_Wind Energy Science, 2017_

## Referee Comment (RC1) · Anonymous Referee #1 · 8 Jan 2018

The paper presents the aeroservoelastic optimization of wind turbine rotor blades based on an established wind turbine optimization code by Zahle. The novelty of the paper is to study the design impact based on a full optimization of smart rotors.

Overall the paper is well written, however, there can be some clarifications made. In chronological order:

p1. The state-of-the-art reference is 8 years old (Barlas and Kuik). p1. Barlas eta al Barlas et al (repetition) p3.25-30 The loads of this design were used as constrained: Do the authors mean the ultimate and the fatigue loads or the loads per wind speed? p3.35 The flaps are controlled with respect to the wind speed. Do the authors mean the hub height inflow or the local sectional wind speed where the flaps operate? p4.5 industrial industrial (repetition) p6.Table 5 The optimizer finds a very high tip speed

for the blades. This can lead to leading edge erosion (and noise, which would be a lesser issue offshore). Did the authors study how limiting the tip speed would change the optimized design? p6.15-17 The authors mention profile drag. For the operation of (discrete) flaps also induced drag of the flaps should be taken into account. A reference to the near wake model would be good. p6. The co-optimization finds higher deflection angles for the flaps without increasing the power. Is there a benefit that is hidden to the reviewer that could point towards using a combined optimization or are the results from the baseline with flap more realistic for actual applications?

---

## Referee Comment (RC2) · Anonymous Referee #2 · 8 Jan 2018

Overall, a very interesting and well written paper. One comment question: the co-design has a much higher flap deflection angle. I'm wondering if this results in larger drag as well? Also, if the flaps were also used for load reduction, and thus varied around this mean angle, would this large mean angle potentially cause issues because the total flap deflection would be quite large?

---

## Author Comment (AC1) · 1 Feb 2018

In response to p1. "The state-of-the-art reference is 8 years old (Barlas and Kuik)." and " p1. Barlas eta al Barlas et al (repetition)" ———————————————————————— ————————————————————————————————————

The following articles will be added:

Johnson, Scott J. et al. "An Overview of Active Load Control Techniques for Wind Turbines with an Emphasis on Microtabs." Wind Energy 13.2-3 (2010): 239–253.

Bernhammer, L. O., G. A. M. van Kuik, and R. De Breuker. "How Far Is Smart Rotor Research and What Steps Need to Be Taken to Build a Full-Scale Prototype?" Journal of Physics. Conference Series 555.1 (2014): 012008.

[Figure]

Chen, Z. J., K. A. Stol, and S. R. Mace. "Wind Turbine Blade Optimisation with Individual Pitch and Trailing Edge Flap Control." Renewable Energy 103 (2017): 750–765.

Smit, Jeroen et al. "Sizing and Control of Trailing Edge Flaps on a Smart Rotor for Maximum Power Generation in Low Fatigue Wind Regimes." Wind Energy 16.4 (2016): 607–624.

Repetition is addressed by adding more articles to the revised manuscript

"p3.25-30 The loads of this design were used as constrained: Do the authors mean the ultimate and the fatigue loads or the loads per wind speed?" —————————————————————————————————————————————————————————————

The fatigue loads are ignored in this optimization due to the added difficulties of optimization with turbulence. This optimization did not consider the use of flaps for fatigue reduction, so it was not considered important to assess quasi-static actuation of flaps. In the revised manuscript, this clarification will be made.

More details on the load cases:

We use a reduced DLB similar to the one that was developed by Pavese:

Pavese, C.; Tibaldi, C.; Larsen, T. J.; Kim, T. & Thomsen, K. Reduced Design Load Basis for Ultimate Blade Loads Estimation in Multidisciplinary Design Optimization Frameworks Journal of Physics: Conference Series, 2016, 753, 062005

For this work the following load cases are used:

DLC 1.2 Regular Operation - Note: without turbulence to acquire steady state response,

DLC 1.3 Extreme Turbulence - Note: this is performed with an extreme gust to mimic the elevated loads seen from turbulence

DLC 2.1 Grid Loss

DLC 2.2b Blade Stuck

DLC 5.1 Emergency Shutdown

DLC 6.1 Parked in extreme winds

DLC 6.2 Parked grid loss

DLC 6.3 Parked with large yaw error

We constrained the root bending loads, tower top thrust, tower bottom moment, then the sectional forces in the blades are used to calculate the ultimate stress in the cross section.

These are typically the load cases that produce the highest loads. The load cases are simplified by not including turbulence. When turbulence is included, local minima occur and the optimization does not progress far. A consequence of this simplifications is that the simulated loads are lower in this simplified DLB than in a Full DLB. This discrepency is resolved by constraining relative increases instead of absolute values. In other words the loads are not allowed to increase. In practice this has proven reliable in a number of projects including the recent re-design of the DTU 10MW that was published in Torque 2016

p3.35 The flaps are controlled with respect to the wind speed. Do the authors mean the hub height inflow or the local sectional wind speed where the flaps operate? ————————————————————————————————————— ————————————————————————————

The flaps are controlled based on the low-pass filtered inflow wind speed. So this would correspond to the hub-height inflow. This will be clarified in the revised manuscript.

p4.5 industrial industrial (repetition) ————————————————————————-

This will be corrected in the revised manuscript

p6.Table 5 The optimizer finds a very high tip speed for the blades. This can lead to leading edge erosion (and noise, which would be a lesser issue offshore). Did the authors study how limiting the tip speed would change the optimized design? —————————————————————————————————————
——————————————————————————————

We agree that the high tip-speed-ratio is unrealistic due to erosion and noise issues. Of course it would be more realistic to have a noise model or an erosion model in the optimization to push the tip speed down. Otherwise the next option is to constrain the tip speed explicitly. However we chose not to include these options to see where the optimization would go. The drag from the large deflections in the flap may push the optimization to lower tip-speed ratios, which we see in the co-design.

In the revised manuscript, we will mention the erosion and noise issues of such a high tip-speed-ratio and state that these considerations were ignored.

p6.15-17 The authors mention profile drag. For the operation of (discrete) flaps also induced drag of the flaps should be taken into account. A reference to the near wake model would be good. ———————————————————————————————
———————————————————————————————————————

Induced drag was not included in this analysis. However, we feel this is a secondary effect and can be safely ignored.

In the revised manuscript it will be stated that the near-wake model of HAWC2 was not used in this work and thus induced drag ignored.

p6. The co-optimization finds higher deflection angles for the flaps without increasing the power. Is there a benefit that is hidden to the reviewer that could point towards using a combined optimization or are the results from the baseline with flap more realistic for actual applications? ———————————————————————————————————
* * *
One of the goals of the study was to investigate whether an "add-on" application was sub-optimal compared to fully integrated design. So we were approaching this research question without any assumptions or insights.

There could be benefits to co-design but they are also hidden from the authors. By the fact that the benefits are not obvious in the optimization results, it indicates that those benefits are smaller than the combined error in the models and the optimization, thus, not revealed with these tools. Thus, for this particular design problem, it seems that a sequential optimization is at least sufficient if not better since it is easier and the blade is still optimal in non-flap applications.

Additional interactions may arise that may lead to a better co-design solutions if the flap is used for more functions and/or more flap design variables (width, length, position, etc.) are exposed. So we think further investigation is needed. We intend on performing further studies, however, we have to address some additional challenges in optimization first.

On page 9, line 10, we state that there are only small benefits to co-design. Thus, it seems nothing should be added to the manuscript to address this comment.

Summary of proposed revisions to the manuscript ————————————————————

- More references will be added to the manuscript

- It will ve be clarified that the ultimate loads were constrained but not fatigue for two reasons. Turbulence effects are difficult to include in optimization and turbulence was not considered important for evaluating the quasi-static actuation of flaps.

- It will be clarified that the flaps were actuated according to the hub-height wind speed.

- repeated word industrial deleted.

- State that the high tip-speed-ratio would lead to greater noise and erosion issues and

that these issues were ignored in the optimization.

- State that induced drag was ignored.

---

## Author Comment (AC2) · 1 Feb 2018

In response to "The co-design has a much higher flap deflection angle. I'm wondering if this results in larger drag as well?": ——————————————————————— ———————————————————————————

Yes, the larger flap angle would increase drag. By the fact that the optimization is choosing large deflections, shows that the load alleviation benefit out-weigh the drag penalty.

In response to "Also, if the flaps were also used for load reduction, and thus varied around this mean angle, would this large mean angle potentially cause issues because the total flap deflection would be quite large?": ———————————————————————

[Figure]

———————————————————————————

The flaps in this case are only used in a quasi-steady way in below rated power operation. Any added load alleviation features will be added in above rated conditions, so there should be no issue with superimposed large flap angles.

Revisions to the manuscript: ——————————-

- In page 9 line 20, it is already mentioned that large flap deflection causes increased drag.

- On page 5 line 11, we mention that the angle was based on a low-passed filtered wind. We will elaborate and state that the flap actuation was quite slow. Furthermore we will comment that that fast flap actuation for fatigue reduction would occur at above rated conditions, where the quasi-static occurs below rated conditions, thus, these two applications do not occur simultaneously.

---

## Author Response (AR1)

**Response to the referees**

Michael McWilliam, *et al.*

**Responses to referee 1:**

**1) Referee 1 comment:** "The state-of-the-art reference is 8 years old (Barlas and Kuik)." and " p1. Barlas eta al Barlas et al (repetition)"

> **Author Response:**
>
> The following articles will be added:
>
> Johnson, Scott J. et al. "An Overview of Active Load Control Techniques for Wind Turbines with an Emphasis on Microtabs." Wind Energy 13.2-3 (2010): 239–253.
>
> Bernhammer, L. O., G. A. M. van Kuik, and R. De Breuker. "How Far Is Smart Rotor Research and What Steps Need to Be Taken to Build a Full-Scale Prototype?" Journal of Physics. Conference Series 555.1 (2014): 012008.
>
> Chen, Z. J., K. A. Stol, and S. R. Mace. "Wind Turbine Blade Optimisation with Individual Pitch and Trailing Edge Flap Control." Renewable Energy 103 (2017): 750–765.
>
> Smit, Jeroen et al. "Sizing and Control of Trailing Edge Flaps on a Smart Rotor for Maximum Power Generation in Low Fatigue Wind Regimes." Wind Energy 16.4 (2016): 607–624.
>
> Repetition is addressed by adding more articles to the revised manuscript
>
> **Revision to the paper:**
>
> The new references can be seen on page 1 line 21, page 2 line 1, page 2 line 9 and page 2 line 10.

**2) Referee 1 comment:** "p3.25-30 The loads of this design were used as constrained: Do the authors mean the ultimate and the fatigue loads or the loads per wind speed?"

> **Author Response:**
>
> The fatigue loads are ignored in this optimization due to the added difficulties of optimization with turbulence. This optimization did not consider the use of flaps for fatigue reduction, so it was not considered important to assess quasi-static actuation of flaps. In the revised manuscript, this clarification will be made.
>
> More details on the load cases:
>
> We use a reduced DLB similar to the one that was developed by Pavese:

Pavese, C.; Tibaldi, C.; Larsen, T. J.; Kim, T. & Thomsen, K. Reduced Design Load Basis for Ultimate Blade Loads Estimation in Multidisciplinary Design Optimization Frameworks Journal of Physics: Conference Series, 2016, 753, 062005

For this work the following load cases are used:

DLC 1.2 Regular Operation - Note: without turbulence to acquire steady state response,
DLC 1.3 Extreme Turbulence - Note: this is performed with an extreme gust to mimic the elevated loads seen from turbulence
DLC 2.1 Grid Loss
DLC 2.2b Blade Stuck
DLC 5.1 Emergency Shutdown
DLC 6.1 Parked in extreme winds
DLC 6.2 Parked grid loss
DLC 6.3 Parked with large yaw error

We constrained the root bending loads, tower top thrust, tower bottom moment, then the sectional forces in the blades are used to calculate the ultimate stress in the cross section.

These are typically the load cases that produce the highest loads. The load cases are simplified by not including turbulence. When turbulence is included, local minima occur and the optimization does not progress far. A consequence of this simplifications is that the simulated loads are lower in this simplified DLB than in a Full DLB. This discrepency is resolved by constraining relative increases instead of absolute values. In other words the loads are not allowed to increase. In practice this has proven reliable in a number of projects including the recent re-design of the DTU 10MW that was published in Torque 2016

**Revision to the paper:**

It will ve be clarified that the ultimate loads were constrained but not fatigue for two reasons. Turbulence effects are difficult to include in optimization and turbulence was not considered important for evaluating the quasi-static actuation of flaps. This can be seen on page 3 line 18.

**3) Referee 1 comment:** "p3.35 The flaps are controlled with respect to the wind speed. Do the authors mean the hub height inflow or the local sectional wind speed where the flaps operate?"

**Author Response:**

The flaps are controlled based on the low-pass filtered inflow wind speed. So this would correspond to the hub-height inflow.

**Revision to the paper:**

This was clarified in the revised paper, page 5 line 7.

**4) Referee 1 comment:** "p4.5 industrial industrial (repetition)"

**Author Response:**

No comment

**Revision to the paper:**

The second "industrial" was removed on page 6 line 3.

**5) Referee 1 comment:** "p6.Table 5 The optimizer finds a very high tip speed for the blades. This can lead to leading edge erosion (and noise, which would be a lesser issue offshore). Did the authors study how limiting the tip speed would change the optimized design?"

**Author Response:**

We agree that the high tip-speed-ratio is unrealistic due to erosion and noise issues. Of course it would be more realistic to have a noise model or an erosion model in the optimization to push the tip speed down. Otherwise the next option is to constrain the tip speed explicitly. However we chose not to include these options to see where the optimization would go. The drag from the large deflections in the flap may push the optimization to lower tip-speed ratios, which we see in the co-design.

**Revision to the paper:**

In the revised manuscript, we will mention the erosion and noise issues of such a high tip-speed-ratio and state that these considerations were ignored. This revision can be seen on page 10 line 7.

**6) Referee 1 comment:** "p6.15-17 The authors mention profile drag. For the operation of (discrete) flaps also induced drag of the flaps should be taken into account. A reference to the near wake model would be good."

**Author Response:**

Induced drag was not included in this analysis. However, we feel this is a secondary effect and can be safely ignored.

**Revision to the paper:**

In the revised manuscript it will be stated that the near-wake model of HAWC2 was not used in this work and thus induced drag ignored. This can be seen on page 3 line 11.

**7) Referee 1 comment:** "p6. The co-optimization finds higher deflection angles for the flaps without increasing the power. Is there a benefit that is hidden to the reviewer that could point towards using a combined optimization or are the results from the baseline with flap more realistic for actual applications?"

**Author Response:**

One of the goals of the study was to investigate whether an "add-on" application was sub-optimal compared to fully integrated design. So we were approaching this research question without any assumptions or insights.

There could be benefits to co-design but they are also hidden from the authors. By the fact that the benefits are not obvious in the optimization results, it indicates that those benefits are smaller than the combined error in the models and the optimization, thus, not revealed with these tools. Thus, for this particular design problem, it seems that a sequential optimization is at least sufficient if not better since it is easier and the blade is still optimal in non-flap applications.

Additional interactions may arise that may lead to a better co-design solutions if the flap is used for more functions and/or more flap design variables (width, length, position, etc.) are exposed. So we think further investigation is needed. We intend on performing further studies, however, we have to address some additional challenges in optimization first.

**Revision to the paper:**

On page 9, line 18, we state that there are only small benefits to co-design. Thus, it seems nothing should be added to the manuscript to address this comment.

**Responses to referee 2:**

**1) Referee 2 comment:** "The co-design has a much higher flap deflection angle. I'm wondering if this results in larger drag as well?":

**Author Response:**

Yes, the larger flap angle would increase drag. By the fact that the optimization is choosing large deflections, shows that the load alleviation benefit out-weigh the drag penalty.

**Revision to the paper:**

In page 10 line 3, it is already mentioned that large flap deflection causes increased drag. So no revision has been made in response to this comment.

**2) Referee 2 comment:** "Also, if the flaps were also used for load reduction, and thus varied around this mean angle, would this large mean angle potentially cause issues because the total flap deflection would be quite large?":

**Author Response:**

The flaps in this case are only used in a quasi-steady way in below rated power operation. Any added load alleviation features will be added in above rated conditions, so there should be no issue with superimposed large flap angles.

**Revision to the paper:**

On page 5 line 7, we mention that the angle was based on a low-passed filtered wind. We will elaborate and state that the flap actuation was quite slow. Furthermore we will comment that that fast flap actuation for fatigue reduction would occur at above rated conditions, where the quasi-static occurs below rated conditions, thus, these two applications do not occur simultaneously. You can see this page 5 line 7